# Comprehensive Characterization of *fucAO* Operon Activation in *Escherichia coli*

**DOI:** 10.3390/ijms25073946

**Published:** 2024-04-02

**Authors:** Zhongge Zhang, Jialu Huo, Juan Velo, Harry Zhou, Alex Flaherty, Milton H. Saier

**Affiliations:** Department of Molecular Biology, School of Biological Sciences, University of California at San Diego, 9500 Gilman Dr, La Jolla, CA 92093-0116, USA; j1huo@ucsd.edu (J.H.); jvelo@ucsd.edu (J.V.); wflahert@ucsd.edu (A.F.)

**Keywords:** *fucAO* operon, operon promoter, Crp, FucR, SrsR, transcriptional activation, transcriptionally silent region, protein-protein interactions

## Abstract

Wildtype *Escherichia coli* cells cannot grow on L-1,2-propanediol, as the *fucAO* operon within the fucose (*fuc*) regulon is thought to be silent in the absence of L-fucose. Little information is available concerning the transcriptional regulation of this operon. Here, we first confirm that *fucAO* operon expression is highly inducible by fucose and is primarily attributable to the upstream operon promoter, while the *fucO* promoter within the 3′-end of *fucA* is weak and uninducible. Using 5′RACE, we identify the actual transcriptional start site (TSS) of the main *fucAO* operon promoter, refuting the originally proposed TSS. Several lines of evidence are provided showing that the *fucAO* locus is within a transcriptionally repressed region on the chromosome. Operon activation is dependent on FucR and Crp but not SrsR. Two Crp-cAMP binding sites previously found in the regulatory region are validated, where the upstream site plays a more critical role than the downstream site in operon activation. Furthermore, two FucR binding sites are identified, where the downstream site near the first Crp site is more important than the upstream site. Operon transcription relies on Crp-cAMP to a greater degree than on FucR. Our data strongly suggest that FucR mainly functions to facilitate the binding of Crp to its upstream site, which in turn activates the *fucAO* promoter by efficiently recruiting RNA polymerase.

## 1. Introduction

1,2-propanediol (PPD) is a common industrial chemical that has been used in the mass production of important commercial products such as biodegradable plastics and polymer resins [1]. PPD is a naturally occurring three-carbon diol, usually derived from the microbial degradation of renewable resources [2]. It is abundantly present in the human gut as well [3]. Many bacteria are capable of growing on PPD [4,5]. However, wildtype *E. coli* K12 strains cannot utilize PPD as a carbon source for growth (PPD^−^), although they harbor the *fucO* gene encoding lactaldehyde:propanediol oxidoreductase (hereafter referred to as propanediol oxidoreductase).

The *fucAO* operon within the *fuc* regulon (Figure 1) encodes an L-fuculose-1-P aldolase (FucA) and a propanediol oxidoreductase (FucO). The aldolase catalyzes the cleavage of L-fuculose-1-P [(a metabolite from the fucose pathway mediated by three enzymes encoded by the *fucPIK* operon: the L-fucose permease (FucP), the L-fucose isomerase (FucI), and the L-fuculose kinase (FucK)], yielding dihydroxyacetone phosphate and L-lactaldehyde. This process is independent of the presence of oxygen.

Under aerobic conditions, L-lactaldehyde is oxidized to L-lactate by aldehyde oxidoreductase (AldA), which is further oxidized to pyruvate by an FAD-dependent dehydrogenase (LldD) prior to entry into central metabolism. The other 3-carbon intermediate, dihydroxyacetone-P, is converted to pyruvate, which feeds into the Krebs (TCA) cycle. However, under anaerobic conditions, L-lactaldehyde (an intermediate metabolite of both the L-fucose and the L-rhamnose catabolic pathways) is reduced to PPD by FucO, and PPD is then excreted from the cell as a waste product [6,7]. Conceivably, in the absence of oxygen, only dihydroxyacetone-P can be used as an energy source for growth. If PPD is present, the same PPD oxidoreductase, FucO, enables the oxidation of PPD back to L-lactaldehyde, which can be converted to L-lactate and then pyruvate prior to entry into the TCA cycle. However, wildtype *E. coli* strains fail to metabolize PPD because the *fucAO* operon is not expressed in the presence of PPD alone. In addition, as an iron (Fe^2+^)-dependent metalloenzyme, FucO is sensitive to oxygen [8,9].

Expression of the *fucAO* operon relies on the Crp-cAMP complex [10]. As part of the *fuc* regulon, the *fucAO* operon is regulated by the regulon regulator FucR as well [11,12,13]. FucR is first activated by its cofactor, fuculose-1-phosphate. FucR, with its cofactor binding, promotes transcription of both *fucPIK* and *fucAO*, and also self-activates. However, to date, the binding sites for Crp and FucR have not been identified within the *fucAO* regulatory region [12]. In addition to Crp and FucR, a recent publication reports that both *fucPIK* and *fucAO* operons are positively regulated by SrsR, a newly identified transcription factor. SrsR is involved in regulating gene expression, mainly during the stationary growth phase [14].

When PPD serves as the sole carbon source, the *fucAO* operon is not induced since *fucR*, encoding FucR, is insufficiently expressed, and no inducer (fuculose-1-phosphate) is synthesized. Although wildtype *E. coli* cells are PPD^−^, PPD^+^ (able to grow on PPD) mutants can readily arise after prolonged incubation with PPD. These PPD^+^ mutants carry an IS5 element inserted upstream of the *fucAO* promoter region. With this insertion, the *fucAO* operon expression is believed to be “constitutive,” while the *fucPIK* operon becomes inactive and non-inducible by L-fucose [13,15,16]. However, it is unknown as to the molecular mechanism by which IS5 activates the *fucAO* operon or whether any host regulatory proteins are involved in IS5 activation of the operon.

Using a predictive computational method, Huerta and Collado-Vides proposed the presence of a σ^70^ promoter (P*_fucAO**.**_*hc), including a −35 element, a −10 element, and the transcriptional start site (+1), upstream of the *fucA* start codon [17]. This putative promoter has an unusually long (165 bp) untranslated region (5′UTR) between +1 and the *fucA* start codon. However, this proposed *fucAO* operon promoter has never been experimentally verified. In addition to the operon promoter, the *fucO* gene is driven by a second promoter (P*_fucO_*) nested in the 3′ end of *fucA* [18]. It is unknown how and to what extent this promoter contributes to *fucO* transcription; it is also unknown if it is inducible by fuculose-1-phosphate like the *fucAO* operon promoter.

Here, we first show that the predicted *fucAO* promoter (P*_fucAO**.**_*hc) is not a true promoter and that the *fucO* promoter (P*_fucO_*) is too weak to cause appreciable induction. Then we identify the actual transcriptional start site (+1) for the main *fucAO* promoter (P*_fucAO_*) (using 5′RACE), which is located at the 63rd nucleotide upstream of the *fucA* start codon. The *fucAO* operon is embedded in a chromosomal region that is transcriptionally repressed. The newly identified promoter (P*_fucAO_*) is highly induced by growth on fucose, and its activity depends on FucR and Crp but not SrsR, either during the logarithmic or the stationary growth phase. Two Crp binding sites and two FucR binding sites were identified and functionally validated. The upstream Crp binding site is more important for operon expression than the downstream site. The second FucR binding site, which is closer to the promoter and is upstream of the Crp binding site, plays a greater role in activating the operon. Although both Crp and FucR are needed for full activation of *fucAO*, we speculate that FucR’s primary function is to facilitate Crp binding, which in turn recruits RNA polymerase and thus initiates transcription.

## 2. Results

### 2.1. The fucAO Operon Activities under Both Non-Inducing and Inducing Conditions

We began our studies by determining the *fucAO* operon activity using a *lacZ* transcriptional reporter (Figure 2A). This reporter strain, ZZ204 (Appendix A), has the *lacZ* gene with its own ribosome binding site (RBS) integrated downstream of *fucO* on the chromosome of strain BW25113 deleted for the native *lacZ* gene, yielding a new operon consisting of *fucA*, *fucO*, and *lacZ*. In this reporter strain, the β-galactosidase activities are proportional to the actual *fucAO* operon expression levels. This strain is still Fuc^+^, as the *fuc* regulon is intact.

To determine basal *fucAO* operon activity under non-inducing conditions, the strain was cultured in minimal M63 medium with glycerol as the sole carbon source. It exhibited 16.6 units of β-galactosidase activity (Figure 2B). This result confirms that the *fucAO* operon is expressed at low levels in the absence of fucose. On the other hand, *fucAO* expression is not “silent” (almost completely shut off) under non-inducing conditions since its expression is readily detectable. When the same strain was grown with fucose as the sole carbon source (inducing conditions), operon expression increased 62-fold to 1023 units of β-galactosidase activity (Figure 2B), confirming that the *fucAO* operon is highly inducible by growth with fucose.

### 2.2. The fucAO Operon Is Activated by Crp and FucR but Not SrsR

To examine how Crp, FucR, and SrsR affect *fucAO* operon expression, the transcriptional operon reporter was transferred to strains with △*crp* (Glp^+^, referring to glycerol positive), △*fucR*, and △*srsR* genetic backgrounds, respectively. These strains were cultured in glycerol minimal medium, and the operon activities are summarized in Figure 2C. A 5-fold further decrease in operon expression was seen in the absence of Crp, although the expression was already weak in wildtype cells. This observation reveals that the global regulator Crp is required to maintain even the low-level expression of the *fucAO* operon under non-inducing conditions. In the absence of FucR, a similar-to-wildtype activity was obtained, consistent with the notion that FucR only activates the *fucAO* operon under inducing conditions (that is, in the presence of fuculose-1-P). As the △*crp* strain and the △*fucR* strain are Fuc^−^, it is impossible to use these deletion strains to determine the effects of Crp and FucR under inducing conditions (Section 2.7 and Section 2.8).

As to SrsR’s effects on *fucAO* transcription, we first measured the operon activity in exponentially growing cells deleted for *srsR*. Under non-inducing conditions, *fucAO* expression was not observably affected as compared with the SrsR^+^ cells (Figure 2C). Under inducing conditions, the loss of SrsR had virtually no effect as well (Figure 2D). We next tested how SrsR impacted *fucAO* expression in the stationary phase since this regulator is reported to be more abundant in this growth stage. When the strains were cultured with glycerol or fucose for extended periods (at an OD_600_ of 3 to 4), no distinct difference in promoter activity was seen between the wildtype and the Δ*srsR* strain (last two columns of Figure 2D).

△*srsR* cells are Fuc*^+^* while △*fucR* or △*crp* cells are Fuc^−^. To see if loss of SrsR reduced growth on fucose, growth rate was measured using fucose minimal media. As shown in Figure 2E, no detectable difference was found when SrsR was deleted. Summarizing these results, we conclude that SrsR is not a significant regulator of *fucAO*, at least under our experimental conditions.

### 2.3. P_fucAO_ and P_fucO_ but Not P_fucAO._hc Contribute to fucAO Operon Expression

As shown in Figure 2A, there are two promoters (P*_fucAO_ and* P*_fucO_*) driving *fucAO* operon expression. To examine the activity of P*_fucAO_* alone, the *lacZ* gene plus its own RBS was integrated downstream of the 10th codon of *fucA*. A stop codon was introduced immediately downstream of *fucA*’s 10th codon (Figure 3A). To determine P*_fucO_* activity, a *rrnB* terminator (T1) was inserted between P*_fucAO_* and *fucA* in the operon reporter strain ZZ204 (Appendix A), blocking transcription from P*_fucAO_* or any other upstream regions (hypothetical promoters) (Figure 3B). To see if the proposed *fucAO* promoter P*_fucAO_*.hc had activity, the same *lacZ* cassette was used to replace the region between −147 and +2372 relative to the *fucA* start site, which included most of the 5′UTR for P*_fucAO_*.hc and the entire *fucAO* operon (Figure 3C).

The activities of these three promoters were measured in cells cultured with glycerol. P*_fucAO_* and P*_fucO_* exhibit nearly identical β-galactosidase activities (about eight units per promoter), and both contribute about 50% of the operon activity (16.6 units). However, P*_fucAO_*.hc alone did not show detectable activity (Figure 3D), suggesting that this predicted promoter may not be an actual promoter.

To further characterize these promoters, it was necessary to examine their activities under inducing conditions. Since the strains carrying these individual promoters cannot grow on fucose due to disruption of *fucAO*, the proposed experiment required the construction of new promoter/*lacZ* reporters at a different chromosomal locus while maintaining the native *fuc* regulon intact (Section 2.6).

To see if P*_fucO_* is inducible like P*_fucAO_*, a new P*_fucO_*/*lacZ* reporter, harboring P*_fucO_* plus *fucO*’s first 10 residues followed by a stop codon, was moved to the *lac* locus while the native *fucAO* was unaltered (Figure 3E). In the presence of fucose, no elevated P*_fucO_* activity was observed (Figure 3F), indicating the *fucO* promoter present in the 3′ end of *fucA* is not inducible. As expected, the loss of FucR had no effect on P*_fucO_* activity (Figure 3G). However, without Crp, the activity decreased eightfold in comparison to the wildtype (Figure 3G). Based on these results, we conclude that the *fucO* promoter (driving *fucO* only), P*_fucO_*, is very weak and is Crp-dependent but FucR-independent.

Similarly, the P*_fucAO_*.hc/*lacZ* cassette shown in Figure 3C was moved to the *lac* locus, leaving the *fuc* regulon intact. P*_fucAO_*.hc refers to the entire *fucPIK*/*fucAO* intergenic region deleted for the 146 bp 3′ end segment (Figure 3H). When this new reporter strain was cultured with glycerol or fucose, P*_fucAO_*.hc still did not display clear promoter activity (Figure 3I). With these results (Figure 3D,I), we conclude that the computer-predicted *fucAO* promoter, P*_fucAO_*.hc, is not an active promoter.

### 2.4. Determination of the Transcriptional Start Site (TSS) for the fucAO Promoter

As described above, the promoter P*_fucO_* appears unimportant for *fucAO* expression, while the formerly predicted *fucAO* promoter P*_fucAO._*hc is not an actual promoter. Our next studies focused on identifying and characterizing the true promoter region (P*_fucAO_*) driving *fucAO* operon transcription, first by determining the transcriptional start site (TSS). To determine the *fucA* TSS, the wildtype strain, BW25113, was cultured in minimal M63 medium with fucose as the sole carbon source. mRNA and cDNA were prepared using the SMARTer^®^ RACE 5′/3′ kit (Takara Bio USA). The 5′ portion of the *fucA* cDNA was amplified using the primer fucA-GSP (Appendix A) that specifically binds to the beginning of *fucA* and the universal primer mix. The PCR products were purified and sequenced using fucA-GSP. Part of the sequencing results are shown in Figure 4A. Instead of an “A” nucleotide (−166 bp from the *fucA* start codon), we observed that the *fucA*’s TSS is a “C” nucleotide, located −63 bp upstream of the *fucA* start codon (Figure 4B).

A σ70 promoter should harbor two well-defined short DNA elements that are situated at about 10 bps and 35 bps upstream of the TSS (the −10 element and the −35 element, respectively) [19]. The consensus sequences are TTGACA (for −35) and TATAAT (for −10), and the distances between these two elements are usually 17 to 18 bps [20]. Based on these features, we proposed two short sequences, “TTGtgA” (−96 to −91 relative to the *fucA* start codon; consensus nucleotides are capitalized) and “aATtAa” (−73 to −68) upstream of the TSS, to be the −35 element and the −10 element, respectively (Figure 4B). This newly identified *fucAO* operon promoter, named P*_fucAO_*, includes the −35 element (TTGtgA), the −10 element (aATtAa), and the transcriptional start site +1(C).

There are four Crp-cAMP binding sites present in the *fucPIK*/*fucAO* intergenic region (Figure 4B). The first two sites (O*_Crp_*0 and O*_Crp_*1) are near the *fucPIK* promoter region and are important for Crp activation of this operon [11,21]. Another two Crp-cAMP binding sites (O*_Crp_*2 and O*_Crp_*3), proposed by Tan et al. [22] and Zheng et al. [23], are situated within the *fucAO* regulatory region. However, these two sites have not been experimentally validated. In a recent study, using a high-throughput binding assay with large amounts of regulatory proteins, Baumgart et al. proposed a FucR binding site (O*_FucR_*2) present in the *fucAO* regulatory region [24]. Based on the consensus sequence, we found a second FucR binding site, O*_FucR_*3. These putative Crp and FucR binding sites have been functionally validated (Section 2.7 and Section 2.8).

### 2.5. The fucAO Operon Is Located within a Transcriptionally Repressed Chromosomal Region

As described above, *fucAO* operon expression is abnormally low (about 16 LacZ units under non-inducing conditions and 1000 units under inducing conditions). To see if such low-level expression is chromosomal loci-dependent, the same P*_fucAO_*/*lacZ* reporter as shown in Figure 3A was moved to the *lac* locus while the native *fucAO* remained unaltered (Figure 5A). Figure 5B shows the chromosomal positions of *fucA* and *lacZ*, which are located symmetrically on both sides of the chromosome at a similar distance (about 993 kb) from *oriC*.

When these reporter strains were cultured with glycerol, P*_fucAO_* at the *lac* locus yielded 495 units of β-galactosidase activity (Figure 5C), which is 66-fold greater than the same promoter at the native *fuc* locus (that is, 7.5 units; Figure 3D), indicating that P*_fucAO_* is a fairly “strong” promoter, even under non-inducing conditions. When cells were grown with fucose, the same promoter at the *lac* locus yielded over 14,500 units of β-galactosidase activity, a 14-fold increase over that obtained at the *fuc* locus (Figure 5D).

We next tested the activities of P*_fucO_* at both locations. P*_fucO_* plus the first 10 codons were substituted for P*_fucAO_* and *fucA*’ in Figure 5A. When cultured with glycerol, P*_fucO_* at the *lac* locus exhibited 5-fold higher activity than at the *fuc* locus (Figure 5E).

To determine if a promoter unrelated to the *fuc* regulon promoters behaved like P*_fucAO_* and P*_fucO_*, the strong constitutive promoter P*_tet_* was used to drive *lacZ* individually at the same *fuc* and *lac* loci. As can be seen in Figure 5F, P*_tet_* located at the *lac* locus exhibited about 3.5 times the activity of the same promoter at the *fuc* locus. Based on the results shown in Figure 5C–F, we conclude that the *fucAO* locus is within a chromosomal region that is transcriptionally repressed. The low-level expression at the *fuc* locus is not due to its distance from the *oriC* site since both loci (*fuc* and *lac*) have similar distances from this site (Figure 1B).

### 2.6. Defining the Minimal fucAO Regulatory Region Having Full Promoter Activity

From this section on, all the promoter activities were measured using the *lacZ* reporter situated at the *lac* locus with the native *fuc* regulon unchanged. Considering the chromosomal location effects on P*_fucAO_* strengths, β-galactosidase activities obtained at the *lac* locus should be divided by the factor “66” (that is, 495/7.5) when tested strains were cultured under non-inducing conditions, or the factor “14” (that is, 14,519/1022) under inducing conditions (Figure 5). With such normalizations, the activities shown should reflect the actual promoter activities obtained when present at the *fuc* locus.

The *fucPIK*/*fucAO* intergenic region is 546 bp in length, and it carries two promoters and multiple potential binding sites for SrsR, Crp, and FucR. To delimit the essential region needed for full P*_fucAO_* activity, various sizes of DNA fragments were deleted from the 5′ end of P*_fucAO_*. As shown in Figure 6A, P*_fucAO_* encompasses the entire 546 bp intergenic region while P*_fucAO__._*V8 only carries 123 bp of the 3′ region. Each of these smaller regions was individually substituted for P*_fucAO_* within the transcriptional P*_fucAO_*-*lacZ* reporter at the *lac* location (Figure 6B).

The promoter activities of these shorter regions were measured using our standard approaches, as shown in Figure 2, Figure 3 and Figure 5. Under the non-inducing conditions (that is, growing the cells in M63+glycerol), P*_fucAO_*.V2 to V7, which are truncated 60 to 380 bp from the 5′ end, still have similar promoter activities as P*_fucAO_*. However, P*_fucAO_*.V8 almost lost its promoter activity, probably due to the loss of one Crp binding site (O*_Crp_*2).

Under inducing conditions (that is, growing the cells with fucose), P*_fucAO_*.V2 to V5 still maintained promoter activities at levels like P*_fucAO_*. P*_fucAO_*.V6 lost about 30% of this activity, likely due to the missing FucR binding site O*_FucR_*2. P*_fucAO_*.V7, which lacked two FucR binding sites, lost most of its promoter activity. As expected, P*_fucAO_*.V8 (missing both FucR binding sites and Crp binding site O*_Crp_*2) (the last column of Figure 6D) had essentially no activity.

For P*_fucAO_* (and versions V2 to V5), over 60-fold increased promoter activities were seen in the inducing cells compared with the noninducing cells (comparing individual columns in Figure 6C,D). This is consistent with operon expression patterns observed under both conditions (Figure 2B). Based on these results, we conclude: (1) The newly identified *fucAO* promoter, P*_fucAO_*, is active and highly inducible; (2) P*_fucAO_*.V5 (−270 to −1 relative to the *fucA* start codon) is the minimal DNA region that still has full promoter activity under both non-inducing and inducing conditions. This region carries the newly identified promoter and all putative FucR and Crp binding sites; (3) The SrsR binding site O*_SrsR_* and the first two Crp binding sites, O*_Crp_*0 and O*_Crp_*1, are not relevant to *fucAO* promoter regulation; (4) FucR binding sites O*_FucR_*2 and O*_FucR_*3 and Crp binding sites O*_Crp_*2 and O*_Crp_*3 appear to be directly involved in P*_fucAO_* activation.

### 2.7. Functionally Validating Crp-cAMP Binding Sites within P_fucAO_

There are four Crp-cAMP binding sites (O*_Crp_*0, O*_Crp_*1, O*_Crp_*2, and O*_Crp_*3) upstream or downstream of the newly identified P*_fucAO_* promoter (Figure 4B and Figure 6A). O*_Crp_*0 and O*_Crp_*1were shown not to be required for P*_fucAO_* activation (Figure 6C,D). Instead, these sites are important for Crp regulation of the *fucPIK* operon [21]. The consensus sequences for Crp binding sites are “aaa**TGTGA**tctaga**TCACA**ttt” (two binding motifs are in bold and capitalized). O*_Crp_*2 with the sequence “tta**gtTGA**accagg**TCACA**aaa” (nucleotides mismatched to the consensus in binding motifs are in the lower case) is located immediately upstream of the −35 element) while O*_Crp_*3 (tag**TGTGA**aaggaa**caACA**tta) is situated in the 5′UTR of P*_fucAO_*. These putative binding sites (O*_Crp_*2 and O*_Crp_*3) have never been functionally validated.

To see if O*_Crp_*2 is important for Crp regulation in P*_fucAO_*, we mutated this site by changing its sequence in P*_fucAO_*.V5 to tta**gtcta**accagg**cgctg**aaa (the altered nucleotides are underlined) (Figure 7A). The promoter activities were measured and recorded in Figure 7. When cultured with glycerol, mutation of O*_Crp_*2 yielded a 14-fold reduced promoter activity, nearly abolishing the basal-level activity detected under noninducing conditions (Figure 7B). When cultured with fucose, the same observation was made with the O*_Crp_*2 mutation, leading to a 350-fold reduction (Figure 7C). These results clearly show that Crp binding site O*_Crp_*2 is a true and essential DNA binding site for Crp-cAMP activation of P*_fucAO_*.

We next mutated the O*_Crp_*3 site by changing its sequence to tag**gcctc**aaggaa**cagat**tta (the altered nucleotides are underlined). With these alterations, a moderate 30% reduction in activity was observed under either non-inducing conditions (Figure 7B) or inducing conditions (Figure 7C). These results show that O*_Crp_*3 is likely another Crp-cAMP binding site, but it is not as important as O*_Crp_*2 in modulating P*_fucAO_* activity. Interestingly, this O*_Crp_*3 site is located within the 5′UTR of P*_fucAO_*, an unusual feature (see Section 3).

### 2.8. Identifying and Validating FucR Binding Sites within P_fucAO_

Using a high-throughput in vitro protein:DNA binding assay, Baumgart et al. (2021) proposed the binding motif for FucR with the consensus sequence: G(A)C(T)C(G)A(C)A(G)A(TC)A(T)**CGG**T(G)**CA**T(A)T(CG), where the bold nucleotides CGG and CA are most conserved [24]. This approach involves protein:DNA binding assays using a large amount of each transcriptional factor of interest and a mixture of genomic DNA fragments. A putative FucR binding site, O*_FucR_*2, was identified (Figure 4B and Figure 6A) [24]. Based on the consensus sequences, we found a second FucR binding site, O*_FucR_*3, upstream of P*_fucAO_* (Figure 4B and Figure 6A). These two putative FucR binding sites had not been functionally characterized.

To determine whether O*_FucR_*2 and O*_FucR_*3 upstream of P*_fucAO_* are true binding sites for FucR, we first mutated O*_FucR_*2 by changing its sequence (CCGAAAACGGTCATT) to CCGAAACGTTTGCTT (the altered nucleotides are underlined) in P*_fucAO_*.V5 (Figure 8A). With such alterations, the promoter activity was negligibly changed when the strains were grown with glycerol as the carbon source, and this was expected since the *fucR* gene was not appreciably transcribed and the cofactor L-fuculose-1-P is not available in the absence of fucose. When the cells were cultured with fucose, the promoter activity was moderately (about 30%) decreased as compared with the wildtype promoter. These results show that O*_FucR_*2 is a true binding site for FucR but does not play a major role in *fucAO* regulation under the conditions used here.

Next, we determined the role of the binding site O*_FucR_*3 in regulating P*_fucAO_*. The sequence (TTAAGAGCGGTCATT) of this site was changed to TTAAGAGGTTCGCTT (the altered nucleotides are underlined; Figure 8A). As expected, the loss of O*_FucR_*3 was essentially without effect on promoter activity when the cells were grown on glycerol (Figure 8B). However, when the cells were grown with fucose, the same mutation dramatically lowered the promoter activity by over 50-fold (Figure 8C). These results show that the binding of cofactor-bound FucR to the second site, O*_FucR_*3, is more important than to the first site, O*_FucR_*2, for the activation of P*_fucAO_*. It is worthwhile to note that the FucR binding site, O*_FucR_*3, is closer to the Crp binding site, O*_Crp_*2, that is essential for Crp activation of P*_fucAO_*.

## 3. Discussion

The *fucAO* operon encodes two important enzymes: a fuculose aldolase and a propanediol oxidoreductase, both of which are involved in the bacterial utilization of L-fucose and L-1,2-propanediol. Contrary to the *fucPIK* operon (the other operon within the *fuc* regulon) and many other sugar metabolic operons in *E. coli*, the *fucAO* operon has not been well studied, especially with respect to its transcriptional regulation. We know that its expression relies on the presence of L-fuculose-1-P, the *fuc* regulon regulator, FucR, and the global activator, Crp. However, little is known as to the actual promoter region and how these regulators activate the *fucAO* operon. In this work, we identified and confirmed the true primary promoter (P*_fucAO_*), driving *fucAO* operon transcription, while refuting the hypothetical promoter (P*_fucAO_*.hc), predicted by a computer program, which has long been used by ecocyc.org. Meanwhile, we found that the *fucO* promoter nested within the *fucA* gene is weak and uninducible, thus being unimportant for operon expression. Two Crp-cAMP binding sites and two FucR binding sites were functionally validated. The second FucR binding site and the first Crp binding site (adjacent to each other) are essential for operon expression, while the other two FucR and Crp sites are needed only for maximal operon expression. Furthermore, we provided evidence that the *fucAO* operon is located within a chromosomal region that is transcriptionally repressed.

Wildtype strains of *E. coli* commonly used in laboratories are unable to utilize propanediol for aerobic growth (PPD^−^) because the *fucAO* operon has long been thought to be “silent” when only PPD is present as a carbon source [12,17]. However, using a sensitive transcriptional operon *lacZ* reporter, we showed that the operon is expressed at a detectable level (over 16 units of β-galactosidase activity yielded from both P*_fucAO_* and P*_fucO_*) in the absence of fucose in the growth medium (i.e., noninducing conditions). Compared with the silent *bglGFB* operon (<1 unit of β-galactosidase activity) [25,26,27], the *fucAO* operon is not “silent.” To aerobically grow on PPD, propanediol oxidoreductase (FucO) is needed in large amounts. This iron (Fe^2+^)-dependent enzyme is only catalytically active under anaerobic conditions due to the loss of Fe^2+^ in the presence of oxygen [10,28]. The negative aerobic growth of wildtype cells on PPD may not only result from the low amount of propanediol oxidoreductase synthesized but also from the catalytic inactivation of the enzyme by oxidation. IS5 insertional mutants are capable of aerobic growth on PPD minimal agar plates, probably due to the presence of a large amount of the oxidoreductase, which is still adequate for cellular growth after being partially inactivated by metal-catalyzed oxidation. In addition, it is known that growing bacterial cells on agar plates can readily generate local anaerobic or micro-anaerobic conditions inside the colony [29,30]. On the other hand, leaky expression might be physiologically relevant, as the cells are always prepared for the presence of fucose or IS5 insertional activation of the *fucAO* operon when PPD is the only carbon source present.

The *fucAO* promoter (Figure 4B), long used by ecocyc.org, was predicted by a computer program [1]. We showed that this promoter has no activity under either inducing or noninducing conditions (Figure 3). The first two Crp binding sites (O*_Crp_*0 and O*_Crp_*1) upstream of the predicted promoter are not relevant to *fucAO* operon activation (Figure 6). The other two Crp binding sites (O*_Crp_*2 and O*_Crp_*3), confirmed to be important for Crp activation of the operon as revealed by this study, are unusually located in the 5′UTR of the predicted promoter used in EcoCyc. In addition, the second FucR binding site, O*_FucR_*3, overlaps the -35 element. Based on these features, it appears impossible for Crp and FucR to activate the *fucAO* operon by binding to these functionally validated sites, consistent with our conclusion that the predicted promoter indicated in EcoCys is not a true promoter.

In this work, we confirmed that FucR and Crp are strong dominant regulators that activate the *fucAO* operon. Meanwhile, our data disprove that SrsR directly regulates the operon in either exponentially growing cells or stationary-phase starving cells. Huerta and Collado-Vides [1] found a SrsR binding site present in the 5′UTR of the *fucPIK* promoter (Figure 4B). However, the presence of this site had not been functionally verified. If this is a valid SrsR binding site, it is unknown how it can elevate the transcription of the *fuc* regulon. Nevertheless, we cannot completely rule out the possibility that SrsR directly or indirectly affects the expression of both or one of the *fucPIK* and/or *fucAO* operons under certain unknown conditions. In the case that SrsR does enhance the *fucPIK* operon expression, it may indirectly impact the *fucAO* operon in the presence of fucose since part of *fucR* expression is driven by the *fucPIK* promoter.

As described above, the *fucAO* operon is not “silent,” even in the absence of fucose. In the current study, we showed that this leaky expression is attributed to the presence of Crp, not FucR, since the loss of Crp abolished expression while the loss of FucR had essentially no effect. In the presence of fucose, it is known that fuculose-1-P bound to FucR dramatically elevates operon expression, but our data indicate such an activation is dependent on Crp. When the upstream Crp binding site was mutated, the same activated FucR failed to promote operon expression (Figure 7), indicating that the presence of Crp bound to P*_fucAO_* is essential for FucR activation of the operon.

Crp regulates at least 180 promoters by binding to one or more 22-bp symmetrical sites with the consensus core half-site TGTGA [23,31,32,33]. Once bound to DNA, Crp directly recruits RNA polymerase to promoters via the formation of the “Crp-αCTD-DNA complex,” thereby initiating transcription [34,35,36,37]. Many Crp-dependent promoters are co-regulated by one or more other factors bound to DNA sites near Crp sites. These coregulators affect Crp binding, either by modifying the local DNA conformation or by increasing the local Crp concentration through direct protein-protein contacts [23,38,39]. In the case of P*_fucAO_*, the binding of FucR likely facilitates the binding of Crp to its sites, which in turn recruits RNA polymerase to the promoter region. Without FucR bound to the nearby upstream region, Crp appears to occupy its sites less efficiently, probably due to the presence of a local DNA structure. Alternatively, DNA-bound FucR might recruit Crp to P*_fucAO_* by direct binding.

Based on these observations, we propose a modulation mechanism as follows: Crp-cAMP is the deterministic regulator, dictating the expression of the *fucAO* operon under both noninducing and inducing conditions. In the absence of fucose, FucR is scarce and has no DNA binding capability due to the lack of the cofactor, fuculose-1-P. In this case, Crp inefficiently binds to P*_fucAO_* due to a local DNA conformation and maintains operon transcription at a basal level. In the presence of fucose, FucR is abundant and activated. When bound to the DNA, FucR facilitates Crp binding to P*_fucAO_* via alteration of the local DNA conformation or direct protein-protein interactions, which in turn recruits RNAP to P*_fucAO_* to initiate transcription.

Our work showed that Crp binding to the upstream O*_Crp_*2 site was vital for *fucAO* transcription, as mutations of this site abolished operon expression. As shown in Figure 4B, a second binding site, O*_Crp_*3, is present downstream of P*_fucAO_*. When this site was mutated, operon expression only slightly decreased, by about 30%. These results indicate that this downstream binding site is not essential for operon transcription, but simultaneous bindings of Crp to both sites are required for maximal transcription. In addition, we showed that Crp binding to this downstream site alone (in the absence of the upstream site) did not affect operon activity, suggesting that this downstream site only plays an accessory role and that its function depends on Crp binding to the upstream site. Another possibility is that Crp is incapable of binding to O*_Crp_*3 in the absence of Crp bound to the upstream site, probably due to an intrinsic DNA structure. It is unknown why Crp can promote *fucAO* expression by binding to a downstream site and how Crp maximizes operon expression by binding to these two sites flanking P*_fucAO_*. In some cases, Crp is able to activate promoters by binding to a downstream 5′UTR region [38,40,41]. Similarly, several other global transcriptional factors, such as ArcA and SoxR, have been reported to activate gene expression by binding to one or more regions downstream of their promoters [42,43]. Alternatively, Crp binding to O*_Crp_*3 located at 5′UTR might have enhanced the mRNA stability, thereby increasing translation efficiency.

Two FucR binding sites were identified and functionally validated in this study. The second site, O*_FucR_*3, is critical, as the loss of this site terminates FucR-mediated induction of the *fucAO* operon. The first site, O*_FucR_*2, located upstream of O*_FucR_*3, plays a secondary role in operon induction, as its mutation only leads to a 30% reduction in operon expression. As shown in Figure 4B, O*_FucR_*3 is adjacent to the upstream Crp binding site O*_Crp_*2, while O*_FucR_*2 is far away. Conceivably bound to O*_FucR_*3, FucR more readily recruits Crp to P*_fucAO_*. Simultaneous bindings of either FucR or Crp to their two sites in P*_fucAO_* are needed in order to further promote operon transcription.

Our study revealed that the same *fucAO* promoter was significantly more active at the *lac* locus than its native locus under both noninducing and inducing conditions. Similar observations were made when using the promoter P*_fucO_* (a weak native promoter) and the promoter P*_tet_* (a strong constitutive promoter). Genes near the origin of replication (*oriC*) generally have higher expression levels due to increased dosages [44,45,46]. However, the different activities for P*_fucAO_* appear not to be due to the distance of these two loci from the *oriC* (Figure 5). The *E. coli* nucleoid is highly compact, and its organization is mediated by DNA supercoiling, macromolecular abundance, and six major nucleoid-associated proteins (NAPs), leading to the formation of multiple topologically isolated loops [47,48,49,50]. Vora et al. reported the presence of transcriptionally silent domains distributed across the chromosome, and these domains overlap with the genomic regions densely bound by NAPs [49]. Furthermore, gene silencing within these regions is predominantly attributed to the abnormally low levels of transcription mediated by DNA structuring proteins (that is, not due to weak promoters) [51]. The *fucAO* operon may be situated within such a transcriptionally silent chromosomal domain. As a major NAP, H-NS has been reported to impede *fucAO* operon expression [52]. H-NS preferentially binds to AT-rich DNA regions. The strong repression might be attributed to the direct binding of H-NS to the *fucAO* regulatory region, which is highly A/T rich (64.3% AT content for the *fucPIK*/*fucAO* intergenic region and 67.5% for the *fucAO* regulatory region from −276 to −34 with respect to +1). H-NS exerts its repressive effect on transcription by reinforcing supercoiled structures of local chromosomal DNA by simultaneously binding to two or multiple target sites and subsequently looping them together, thus trapping RNAP at or excluding it from the promoter [27,53,54,55]. This could be the case for H-NS repression of the *fucAO* operon. More studies are needed to examine how the *fucAO* locus is transcriptionally silenced.

In conclusion, we have provided a detailed study on transcriptional regulation of the *fucAO* operon by identifying and validating the true primary promoter and the binding sites for two major activators, Crp and FucR. Operon expression is exclusively dependent on Crp, while FucR, once activated and bound to DNA, appears to favor the binding of Crp to the promoter, thus recruiting RNA polymerase to initiate transcription. Further studies are needed to answer why Crp must bind to both upstream and downstream sites to maximize operon expression, to see if Crp binding to the downstream O*_Crp_*3 site promotes mRNA stability, and to examine the molecular mechanisms by which Crp and FucR coordinate each other’s effects in activating the *fucAO* promoter. Furthermore, it will be of interest to investigate how the *fucAO* operon is silenced at its native locus since the promoter is more active at another chromosomal location.

## 4. Materials and Methods

### 4.1. E. coli Strains and Growth Conditions

Bacterial strains and plasmids used in this study are described in Appendix A. All test strains were derived from *E. coli* K12 strain BW25113 [56]. Using the Lambda-Red recombination system [56], the chromosomal region carrying the *lacI*, *lacZ*, and *lacY* genes was first replaced by a kanamycin marker (*km^r^*) that was subsequently flipped out by pCP20, yielding strain ZZ200. Similarly, the *fucR* gene and the *srsR* gene were deleted from ZZ200, yielding strains ZZ201 and ΔZZ202, respectively. The *lacIZY* mutation was transferred to strain Δ*crp* Glp^+^ (able to utilize glycerol) [57], yielding strain ZZ203. The primers used in this study are listed in Appendix A.

To genetically modify *E. coli* strains, they were routinely cultured in LB media with one or two appropriate antibiotics at 30 °C or 37 °C. To measure the *fucAO* promoter or operon activities, test strains were cultured in M63 minimal media with either 0.5% (*w*/*v*) glycerol or 0.5% (*w*/*v*) L-fucose as the carbon source. The 10× M63 salt solution contains 15 mM (NH_4_)_2_SO_4_, 100 mM KH_2_PO_4_, and 0.02 mM FeSO_4_·7H_2_O. After diluting to 1× M63 medium, it was supplemented with 10**^−^**^4^% thiamine (*w*/*v*) and 1.7 mM MgSO_4_. When necessary, ampicillin, kanamycin, and chloramphenicol were added to the media at 100 μg/mL, 25 μg/mL, and 10 μg/mL, respectively.

### 4.2. Construction of the fucAO Operon Transcriptional LacZ Reporter at the fuc Locus

The *lacZ* structural gene plus its upstream ribosome binding site (RBS), which is CACAGGAAACAGCT, was amplified from the genomic DNA of strain MG1655. A *cat* gene (encoding chloramphenicol resistance) with its constitutive promoter (P*_cat_*) was amplified from pZA31 [58]. Using fusion PCR, these two fragments were combined by fusing the 3′ end of *lacZ* to the 5′ end of P*_cat_*, yielding a fusion fragment “*lacZ*:*cat*” (note: *lacZ* has its own RBS and *cat* has its own promoter). Using the Lambda-Red system, the “*lacZ*:*cat*” cassette was chromosomally integrated downstream of *fucO* in strain ZZ200 to substitute for a 15-bp region (tgatgtgataatgcc) between the 5th and the 22nd nucleotides relative to the *fucO* stop codon. This yielded the transcriptional reporter strain ZZ204, in which *fucA*, *fucO*, and *lacZ* form an operon driven by the *fucAO* operon promoter (P*_fucAO_*) upstream of the *fucA* gene. In addition, among this “*fucA*:*fucO*:*lacZ*” operon reporter, genes *fucO* and *lacZ* are driven by the *fucO* promoter (P*_fucO_*) located in the 3′ end of *fucA* [18] as well (Figure 2A). As expected, the reporter strain remains fucose-positive (Fuc^+^) since the *fuc* regulon was unchanged.

To test the dependence of *fucAO* expression on Crp, FucR, and SrsR, the *fucAO* transcriptional operon reporter was individually transferred to deletion mutants Δ*crp* Glp^+^ [57], Δ*fucR*, and Δ*srsR*, yielding strains ZZ205, ZZ206, and ZZ207, respectively.

### 4.3. Construction of Chromosomal Promoter-LacZ Reporters at the fuc Locus

To determine the activity of the main promoter (P*_fucAO_*) driving expression of the *fucAO* operon, the “*lacZ*:*cat*” cassette (note: *lacZ* has its own RBS) was moved immediately downstream of the 10th codon of *fucA* (referred to as *fucA*’) while the remaining part of *fucAO* was deleted. A stop codon, TAA, was introduced between *fucA*’ and *lacZ*. This yielded strain ZZ208, in which *fucA*’ and *lacZ* form an operon driven by P*_fucAO_* alone at the native *fuc* locus (Figure 3A).

To test the activity of P*_fucO_* alone, a *rrnB* terminator (T1) was inserted between P*_fucAO_* and *fucA*. Briefly, the region for *km^r^* and T1 was amplified from plasmid pKDT [26]. The PCR products were gel purified and subsequently integrated immediately upstream of the beginning of *fucA* in strain ZZ204. The *km^r^* gene was flipped out, yielding strain ZZ209, in which *fucO* and *lacZ* are driven only by P*_fucO_* since P*_fucAO_* is blocked by the *rrnB* terminator T1 (Figure 3B).

To see if the proposed *fucAO* promoter P*_fucAO_*.hc is active, the same “*lacZ*:*cat*” cassette was substituted for the region of −147 to +2372 with respect to the *fucA* start site, including most parts of the 5′UTR and the entire *fucAO* operon. This yielded strain ZZ210, in which P*_fucAO_*.hc alone drives *lacZ* at the *fuc* locus (Figure 3C). It is worthwhile to note that these three promoter *lacZ* reporters are Fuc^−^ due to the lack of the *fucAO* operon.

### 4.4. Construction of Chromosomal Promoter-LacZ Reporters at the lac Locus

To further characterize these promoters, the same three promoter *lacZ* transcriptional reporters described in Section 4.3 were individually moved to the *lac* locus while leaving the native *fuc* regulon intact. With these strains, promoter activities can be examined under both noninducing and inducing conditions. To construct P*_fucAO_* driving *lacZ* at the *lac* locus, the cassette “P*_fucAO_*-*fucA’-lacZ*” shown in Figure 3A was moved to the *lac* locus. Briefly, the entire *fucPIK*/*fucAO* intergenic region plus the first 10 codons of *fucA* (that is, −546 to +30 with respect to the *fucA* start site) followed by a stop codon was cloned into pKDT, yielding pKDT-P*_fucAO_* (Appendix A). The “*km*^r^:T:P*_fucAO_*” cassette (containing the first 10 *fucA* codons and a stop codon) was integrated upstream of the 14th nucleotide with respect to the *lacZ* translational start site in strain MG1655 deleted for *lacY* [59], replacing the *lacI* gene and the *lacZ* promoter. The reporter was transferred to BW25113, yielding strain ZZ211, in which P*_fucAO_* alone drives *lacZ* transcription at the *lac* locus while the native *fuc* regulon is unchanged (Figure 5A).

Similarly, seven shorter promoter versions (−480 to +30, −377 to +30, −339 to +30, −270 to +30, −206 to +30, −166 to +30, and −123 to +30 relative to the *fucA* start site) with various truncations from the 5′ end of P*_fucAO_* were individually substituted for P*_fucAO_* in the P*_fucAO_*-*lacZ* reporter cassette at the *lac* locus, yielding strains ZZ212, ZZ213, ZZ214, ZZ215, ZZ216, ZZ217, and ZZ218, respectively. These resultant strains harbor promoters P*_AO_*.V2, P*_AO_*.V3, P*_AO_*.V4, P*_AO_*.V5, P*_AO_*.V6, P*_AO_*.V7, and P*_AO_*.V8 that individually drive *lacZ* at the *lac* locus (Figure 6A,B).

To construct P*_fucO_* driving *lacZ* at the *lac* locus, the region of −449 to +30 with respect to the *fucA* start site (carrying P*_fucO_* and the first 10 codons of *fucO* followed by a stop codon) was cloned into pKDT, yielding pKDT_P*_fucO_* (Appendix A). The “*km*^r^:T:P*_fucO_*” cassette was inserted into the same position as ZZ211 in strain MG1655Δ*lacY* and subsequently transferred to BW25113, yielding strain ZZ219, in which P*_fucO_* alone drives *lacZ* transcription at the *lac* locus (Figure 3E).

To construct P*_fucAO_*_._hc driving *lacZ* at the *lac* locus, the region from −546 to −147 relative to the *fucA* translational site was amplified from pKDT_P*_fucAO_* and then inserted into the same *lac* position as for P*_fucAO_* in ZZ211. The reporter was transferred to BW25113, yielding strain ZZ220, in which P*_fucAO_*_._hc alone drives *lacZ* at the *lac* locus.

### 4.5. Determining Transcriptional Start Sites Using SMARTer^®^ RACE 5′/3′ Kit

To prepare total RNA, strain BW25113 was shaken at 37 °C in M63 minimal media with 0.5% fucose as the sole carbon source. At OD_600_ of about 1.0, a 600 μL culture was vortexed with 1.2 mL RNAprotectTM Bacteria Reagent (Qiagen, Hilden, Germany) in a 2.0 mL microcentrifuge tube. After 5 min of incubation at room temperature, the mixture was centrifuged at 5000 rpm for 10 min, and the pellet was air dried for 5 min before being frozen at −20 °C. A NucleoSpin^®^ RNA Kit (Takara Bio, San Jose, USA) was used to extract total RNA from the frozen cell pellet. The pellet was first lysed with lysozyme (1 mg/mL) and subsequently bound to the NucleoSpin Filter. The NucleoSpin filter was desalted, treated with the provided rDNase (to remove residual DNA), washed, and dried prior to RNA elution with RNase-free deionized water. The eluted total RNA samples were stored at −80 °C, and the absorbance ratios 260/280 and 260/230 of the eluted RNA samples were measured using a NanoDrop 1000 (Thermo Fisher, Waltham, USA) to ensure RNA purity.

mRNA was extracted using a MICROB*Express*^TM^ Bacterial mRNA Purification Kit (Invitrogen). The total RNA sample was thawed slowly on ice, mixed with 100% ethanol, and centrifuged (12,000 rpm for 30 min) at 4 °C. The resulting RNA pellet was washed three times using 70% ethanol, air dried, and dissolved in 15 μL TE buffer (containing 10 mM Tris-HCl and 1 mM EDTA at pH 8.0). RNA was then introduced to the provided binding buffer with Capture Oligo Mix. The mixture was incubated at 70 °C for 10 min and 37 °C for 30 min to denature the 16S and 23S rRNAs and facilitate hybridization of the rRNAs to capture oligonucleotides. The mixture was combined with the MagBeads and incubated for 15 min at 37 °C to allow the MagBeads to anneal to the hybridized oligonucleotides bound to the rRNA. The MagBead slurry was then placed in a magnetic stand to draw the MagBeads from the solution, leaving supernatant. The mRNA present in the supernatant was precipitated using 5 mg/mL glycogen, 3M sodium acetate, and 100% ethanol. After centrifugation (13 K rpm, 30 min), the mRNA pellet was washed (with 70% ethanol), air dried briefly, and dissolved in nuclease-free deionized water. The concentration and purity of the mRNA samples were determined by measuring their absorbance ratios 260/280 and 260/230 before storing them at −80 °C.

5′RACE was performed using the SMARTer^®^ RACE 5′/3′ kit (Takara Bio USA). First, to synthesize first-strand cDNA, the extracted mRNA sample was combined with a random hexamer mixture that binds to the mRNA. The mixture was incubated at 72 °C for 3 min and then 42 °C for 2 min. A buffer containing RNase inhibitor, Reverse Transcriptase, and SMARTer II Oligonucleotide (all provided) was added to the mixture which was subsequently incubated at 42 °C for 90 min and then 70 °C for 10 min. The resulting mixture (first-strand cDNA) was diluted with tricine-EDTA buffer. After dilution, the first-strand cDNA was combined with a PCR master mix, 5′ gene-specific primer (that is, GSP-fucA-R) (Appendix A), and the universal primer mix for amplification. PCR products (that is, amplified cDNA) were purified by gel electrophoresis, and the purified cDNA was subsequently submitted for sequencing. The first nucleotide immediately downstream of the SMARTer II Oligonucleotide sequence is the transcriptional start site (+1) of the target gene.

### 4.6. Alteration of Crp and FucR Binding Sites within the fucAO Regulatory Region

Crp and FucR are the primary regulators activating *fucAO* operon expression. There are two Crp binding sites (O*_Crp_*2 and O*_Crp_*3) and two FucR binding sites (O*_FucR_*2 and O*_FucR_*3) identified within the *fucAO* regulatory region (Figure 4B and Figure 6A). Thus far, these binding sites have not been validated. The “P*_AO_*.V5-*lacZ*” reporter cassette harbored in strain ZZ214 was employed for examining the functions of these binding sites, as the promoter P*_AO_*.V5 carries all the binding sites and has full promoter activity under both noninducing and inducing conditions. Fusion PCR was used to mutate these sites by changing some key nucleotides within the binding motifs. P*_AO_*.V5 was divided into two separate fragments with a short (about 30 bp) overlapped region in between. The nucleotides to be altered were included in the overlapped region. These two fragments were ligated by fusion PCR, yielding a fusion product with the desired altered nucleotides on the binding sites.

The consensus sequence for Crp binding sites is “aaa**TGTGA**tctaga**TCACA**ttt” (two binding motifs are in bold and capitalized). To mutate O*_Crp_*2 with the sequence “ttagt**TGA**accagg**TCACA**aaa” (the motif nucleotides matching the consensus ones are capitalized), 8 nucleotides within two binding motifs were replaced by other nucleotides, resulting in the sequence “ttagtctaaccaggcgctgaaa” (the altered bases are underlined). To mutate O*_Crp_*3, its sequence “tagTGTGAaaggaacaACAtta” was changed to “taggcctcaaggaacagattta,” in which 8 nucleotides were altered. These two modified P*_AO_*.V5 versions with mutated O*_Crp_*2 and O*_Crp_*3 were substituted for P*_AO_*.V5, first in plasmid pKDT-P*_AO_*.V5 and then in strain ZZ215, yielding strains ZZ221 and ZZ222, in which *lacZ* is exclusively driven by P*_AO_*.V5 with mutated O*_Crp_*2 (for ZZ221) or O*_Crp_*3 (for ZZ222).

The proposed binding motif for FucR has the consensus sequence “G(A)C(T)C(G)A(C)A(G)A(TC)A(T)**CGG**T(G)**CA**T(A)T(CG)”, where the bold nucleotides CGG and CA are most conserved [24]. Based on this consensus sequence, two FucR binding sites (O*_FucR_*2 and O*_FucR_*3) were identified with respective sequences as “ccgaaaaCGGtCAtt” and “ttaagagCGGtCAtt” (Figure 4B). Using the same strategy as above, these binding sites were altered by respectively changing their sequences to ccgaaacgtttgctt and ttaagaggttcgctt (the altered nucleotides are underlined) in P*_AO_*.V5. These modified P*_AO_*.V5 with the mutated FucR binding sites were substituted for P*_AO_*.V5 in strain ZZ215, yielding strains ZZ223 and ZZ224, in which *lacZ* is exclusively driven by P*_AO_*.V5 with mutated O*_FucR_*2 (for ZZ223) or O*_FucR_*3 (for ZZ224).

### 4.7. β-Galactosidase (LacZ) Activity Assay

To prepare samples for β-galactosidase (LacZ) assay, a fresh colony from the reporter strain of interest was cultured in 5 mL of LB media at 37 °C with shaking for about 6 h. 30 μL of the culture was transferred to another tube containing 3 mL of collection media. In this study, the collection media used were M63 minimal media with 0.5% glycerol (noninducing conditions) and M63 minimal media with 0.5% fucose (inducing conditions). The M63 culture was then left to grow overnight at 37 °C with shaking. The next day, a specific amount of overnight culture (preculture) was inoculated into 5 mL of the same collection medium to OD_600_ of 0.02, and the new culture was grown at 37 °C with shaking. During the exponential growth phase, at least four samples were collected at OD_600_ between 0.2 and 1.0. Collected samples were immediately frozen at −20 °C prior to the assay.

To measure the β-galactosidase (LacZ) activities, the previously collected samples were first thawed to room temperature. Then 200 μL of sample, 800 μL of Z-Buffer, and 25 μL of chloroform were combined in a small glass tube and vortexed twice at 10 s each. The sample tubes were placed into a water bath incubator and warmed to 37 °C. To initiate the reaction, 200 μL of prewarmed o-nitrophenyl-β-D-galactopyranoside (β-ONPG) at 4 mg/mL was added to each sample. After a yellow color was visibly developed, 0.5 mL of 1M sodium carbonate was added to each sample and vortexed to stop the reaction. The reaction mixture was appropriately diluted and then centrifuged for 2.5 min at 15,000 rpm. Absorbance values of the prepared reaction mixtures were measured at 420 nm and 550 nm. The β-galactosidase activity (Miller units) for each sample was then calculated using the formula: [1000 × (OD_420_ − 1.75 × OD_550_) x Dilution factor]/[Time of reaction (min) × Volume of sample (mL)] [60]. The slope of LacZ activities in Miller units versus ≥4 collected OD_600_ values represented the reporter strain activity. The final β-galactosidase activity for each strain was the average of at least three repeats (that is, at least 12 samples per strain).

### 4.8. Growth Rate Measurement

One fresh colony of the test strain was cultured in LB with shaking for 8 h. 20 μL of the culture was transferred to 3 mL M63 minimal medium with 0.5% fucose. After overnight growth with shaking at 37 °C, an appropriate amount of the culture was inoculated into 5 mL of the same M63 + fucose medium within a glass tube at the initial OD_600_ of 0.01. The tube was shaken (250 rpm) at 37 °C. In the range of OD_600_ from 0.1 to 1, five or more samples were taken at various time intervals for OD_600_ measurements. The slope of OD_600_ in log values versus time (minutes) represents the growth rate (that is, time per doubling).

### 4.9. Statistical Analysis

All β-galactosidase activity data are expressed as mean *±* standard deviation (SD). Statistical significance was tested by either two-sample *t*-test (for 2 treatments) or 1-way ANOVA followed by Tukey Kramer’s post hoc test (for ≥3 treatments). All figures and β-galactosidase activities were generated using Microsoft Excel (Version 16.66.1) or RStudio (Version 2023.12.0 + 369 “Ocean Storm” Release for Windows). Details of the statistical tests used are indicated in the figure legends. Sample size details are described in Section 4.7 and the legend of Figure 2. Between each of two treatments shown in figures dealing with *≥*3 treatments, different lowercase letters marked above the bar graphs represent statistically significant differences at *p*-values < 0.05, while the same letters represent no significant differences at *p*-values > 0.05.

## Figures and Tables

**Figure 1 ijms-25-03946-f001:**
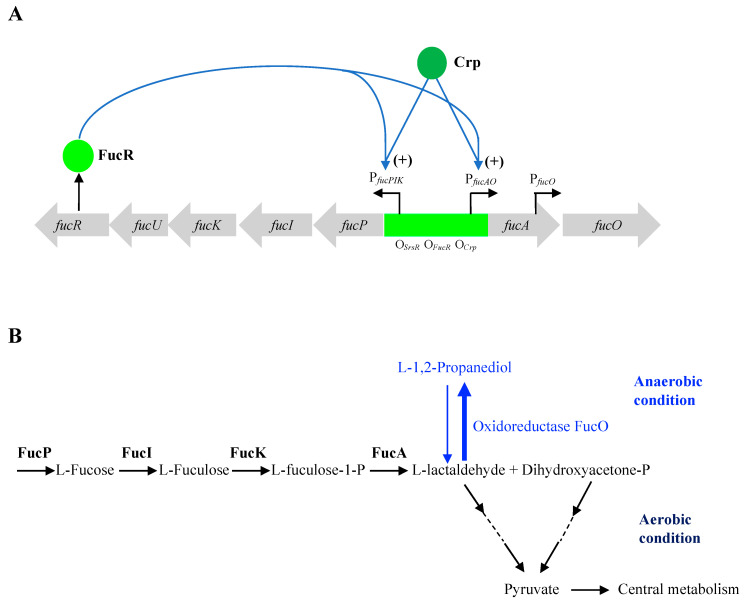
The *fuc* regulon and the L-1,2-propandiol pathway. (**A**) Organization of the *fuc* regulon and its regulation. The *fuc* regulon consists of the *fucPIK* operon and the *fucAO* operon. Both operons have been reported to be positively regulated by Crp, FucR, and SrsR, although some of their binding sites have not been identified or validated. The *fucO* gene is driven by the operon promoter P*_fucAO_* and a promoter P*_fucO_* within the *fucA* gene. The *fucAO* operon is thought to be silent in wildtype cells. (**B**) Diagram showing the pathway of L-1,2-propandiol (PPD) utilization in *E. coli*. Wildtype cells cannot aerobically grow on PPD because the *fucO* gene, encoding an oxidoreductase, is expressed at very low levels, and the oxidoreductase is oxygen-sensitive. The oxidoreductase (FucO) catalyzes the reversible conversion between PPD and L-lactaldehyde, an intermediate metabolite generated in the fucose pathway. When the oxidoreductase is produced in a large amount upon IS5 insertion upstream of P*_fucAO_*, the cells can grow aerobically on PPD.

**Figure 2 ijms-25-03946-f002:**
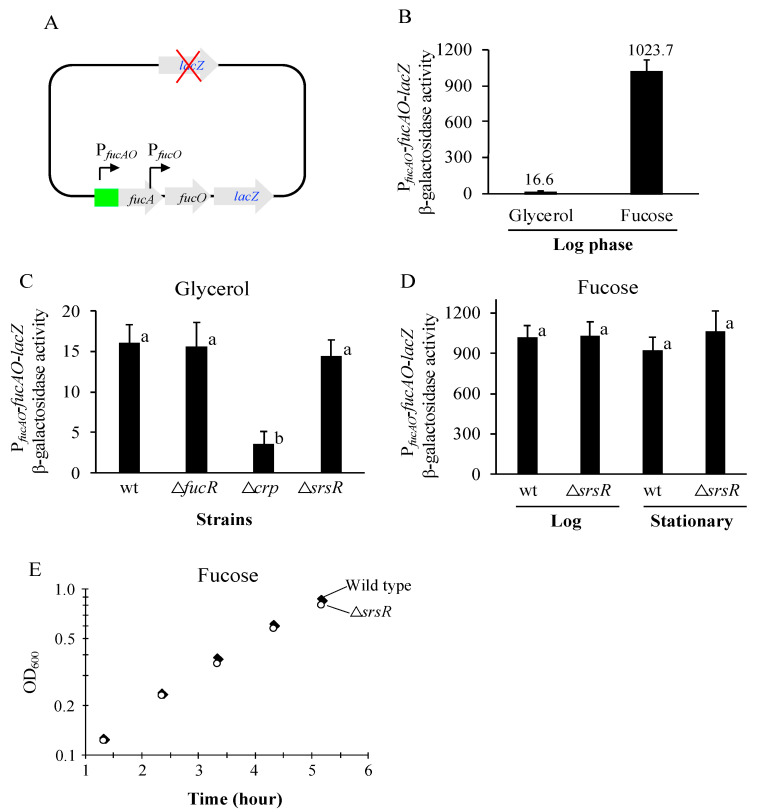
*fucAO* operon activities in wildtype cells and their various genetic backgrounds**.** Test strains were cultured in M63 minimal media with shaking at 37 °C. For each β-galactosidase assay (that is, one repeat), ≥4 samples were collected at OD_600_ values of 0.2 to 1.0 during the exponential growth period. At least three repeats were conducted for each treatment (that is, one strain per growth condition). For each treatment, *n* ≥ 12 was used (Section 4.7). Between each of the two treatments, different lowercase letters marked above the bar graphs represent statistically significant differences at *p*-values < 0.05, while the same letters represent no significant differences at *p*-values > 0.05. All these criteria apply to all subsequent β-galactosidase activity figures. Bacterial samples were subjected to β-galactosidase assays as described in Section 4, and the enzyme activities were calculated using the equation [(OD_420_ − 1.75 × OD_550_)/(sample volume in mL × time in min)] × 1000. For each treatment, the slope of OD_600_ values versus β-galactosidase activities was defined as the promoter or the operon activity. (**A**) Diagram showing the *lacZ* transcriptional reporter for the *fucAO* operon (P*_fucAO_*-*fucAO*-*lacZ*). The red cross denotes the deletion of the native *lacZ* gene. The *lacZ* gene plus its RBS were integrated downstream of the *fucO* gene within the native *fucAO* operon, while the native *lacZ* was deleted. (**B**) The *fucAO* operon activities under noninducing (glycerol) and inducing (fucose) conditions (*p* < 0.05). (**C**) Effects of deleting *fucR*, *crp*, or *srsR* on *fucAO* operon activities in cells growing with glycerol (ANOVA *p* < 0.001). Δ*crp* exhibited significantly lower operon activities than wt (*p* < 0.001), Δ*fucR* (*p* < 0.05), and Δ*srsR* (*p* < 0.01). wt, Δ*fucR*, and Δ*srsR* had similar activities (all *p*-values > 0.05). (**D**) Effects of deleting *srsR* on *fucAO* operon activities in exponentially growing cells and stationary phase cells (ANOVA *p* > 0.05). Strains were cultured with fucose. Samples were collected for β-galactosidase assays at OD_600_ values of 0.2 to 0.8 (exponentially growing cells) and OD_600_ values of 3 to 4 (stationary phase cells). No significant operon activities were observed between these two strains in either logarithmic or stationary growth phases (all *p* > 0.05). (**E**) Growth rate measurements for wildtype and Δ*srsR* mutant strains grown in fucose-minimal media.

**Figure 3 ijms-25-03946-f003:**
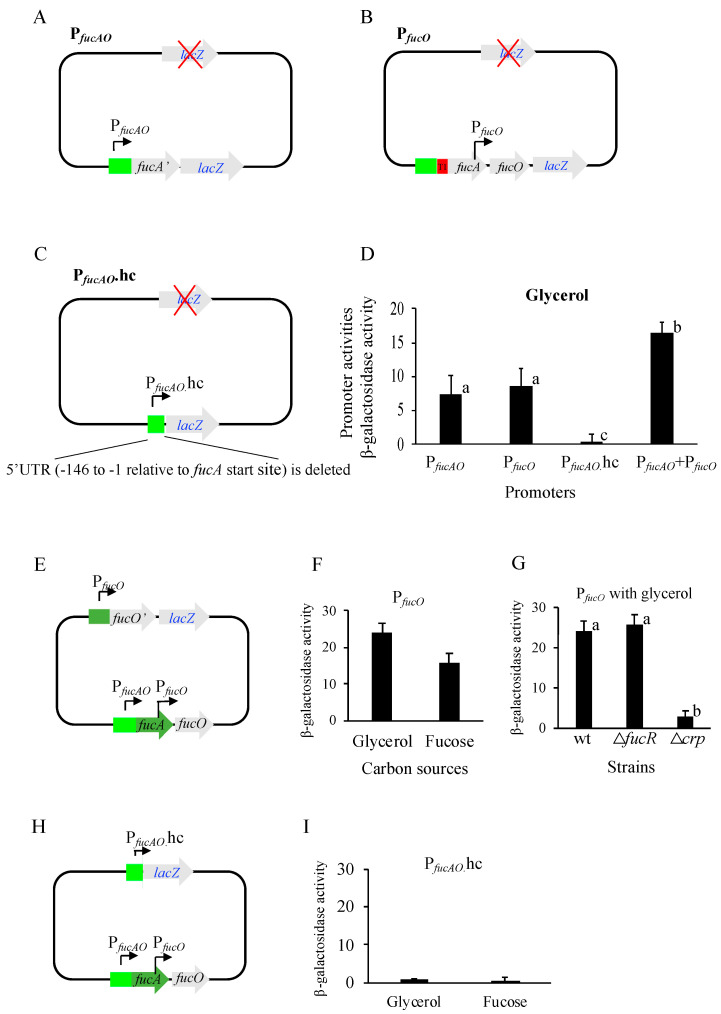
The activities of promoters P*_fucAO_*, P*_fucO_*, and P*_fucA_*_O_.hc. P*_fucAO_* refers to the primary operon promoter upstream of the *fucA* gene, which drives the expression of the *fucAO* operon. P*_fucO_* within the 3′ end of *fucA* drives the expression of *fucO* only. P*_fucA_*_O_.hc refers to the proposed *fucAO* promoter predicted by a computer program. The red cross in (**A**–**C**) refers to the deletion of the native *lacZ* gene. (**A**) Diagram showing P*_fucAO_* alone driving *lac*Z. *fucA*’ refers to the first 10 *fucA* codons, followed by a stop codon. (**B**) Diagram showing P*_fucO_* alone driving *lac*Z. An *rrnB* terminator T1 was inserted between P*_fucAO_* and *fucA*, blocking transcription from P*_fucAO_*. (**C**) Diagram showing P*_fucAO._*hc alone driving *lac*Z. Part of the 5′UTR (−146 to −1 relative to the *fucA* start site) plus *fucA*’ was deleted from the P*_fucAO_* reporter shown in (**A**). (**D**) The promoter activities in noninducing cells grown with glycerol as the carbon source (ANOVA *p* < 0.0001). Different lowercase letters labeled on the bars refer to significant differences. P*_fucAO_* and P*_fucO_* had similar activities (*p* > 0.05). P*_fucA_*_O_.hc had significantly lower activities than P*_fucAO_* (*p* < 0.05), P*_fucO_* (*p* < 0.01), and P*_fucAO_*+P*_fucO_* (*p* < 0.0001). P*_fucAO_*+P*_fucO_* exhibited significantly higher activities than P*_fucAO_* (*p* < 0.01) and P*_fucO_* (*p* < 0.01). (**E**) Diagram showing P*_fucO_* alone driving *lac*Z at the *lac* locus. P*_fucO_* plus the first 10 *fucO* codons replaced the *lacZ* promoter except for the RBS at the *lac* locus. *fucO*’ refers to the first 10 *fucO* codons, followed by a stop codon. (**F**) The P*_fucO_* promoter activities in noninducing cells and inducing cells (*p* > 0.05). (**G**) Effects of deleting *fucR* and *crp* on P*_fucO_* promoter activities in noninducing cells (ANOVA *p* < 0.05). Different lowercase letters labeled on the bars refer to significant differences. Δ*crp* exhibited significantly lower operon activity than wt (*p* < 0.05) and Δ*fucR* (*p* < 0.05). Δ*fucR* and wt had similar activities (*p* > 0.05). (**H**) Diagram showing P*_fucAO_*_._hc alone driving *lac*Z at the *lac* locus. P*_fucAO_*_._hc and *lacZ* are the same, as shown in (**C**). (**I**) The P*_fucAO_*_._hc promoter activities in noninducing cells and inducing cells (*p* > 0.05).

**Figure 4 ijms-25-03946-f004:**
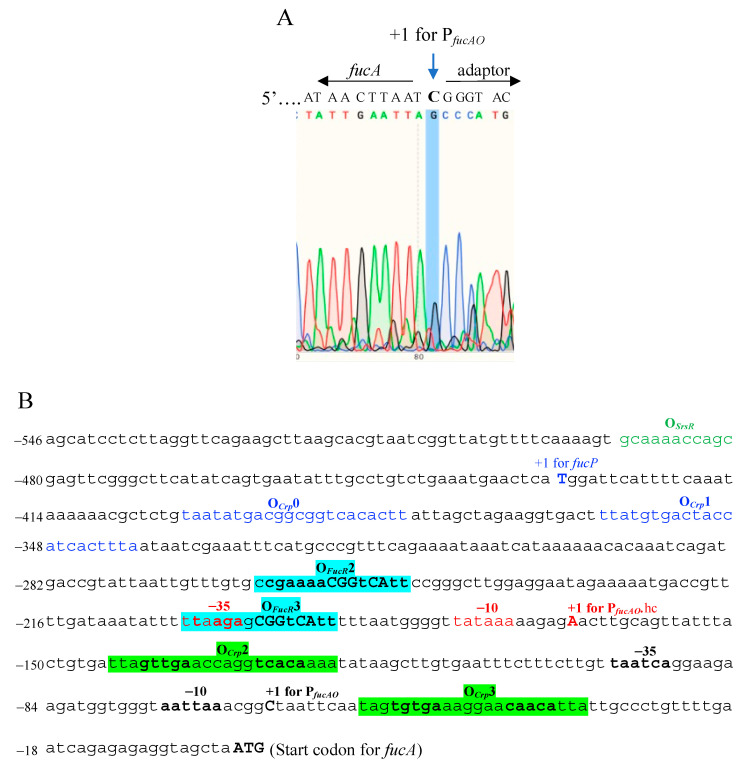
Determining the transcriptional start site (TSS) for P*_fucAO_* using 5′RACE. (**A**) Sequencing chromatogram of a part of the *fucA* cDNA showing the TSS. The capital “C” is the transcriptional start site determined by 5′RACE. The sequence on the left side of the TSS is the beginning sequence of the *fucA* cDNA, while the sequence on the right side is from a sequencing adaptor provided within the Takara Bio kit (San Jose, CA, USA). (**B**) The entire *fucPIK*/*fucAO* intergenic region, showing the newly identified P*_fucAO_* together with both Crp and FucR binding sites. The transcriptional start site (+1), the −10 element, and the −35 element are in bold. Two Crp binding sites (O*_Crp_*2 and O*_Crp_*3) upstream and downstream of P*_fucAO_*, respectively, are highlighted in green, while two FucR binding sites (O*_FucR_*2 and O*_FucR_*3) are highlighted in cyan. The predicted *fucAO* promoter regions are in red. The TSS for *fucP* is capitalized and in blue. Two Crp binding sites (O*_Crp_*0 and O*_Crp_*1), associated with the *fucP* promoter, are in blue. The SrsR binding site is in green.

**Figure 5 ijms-25-03946-f005:**
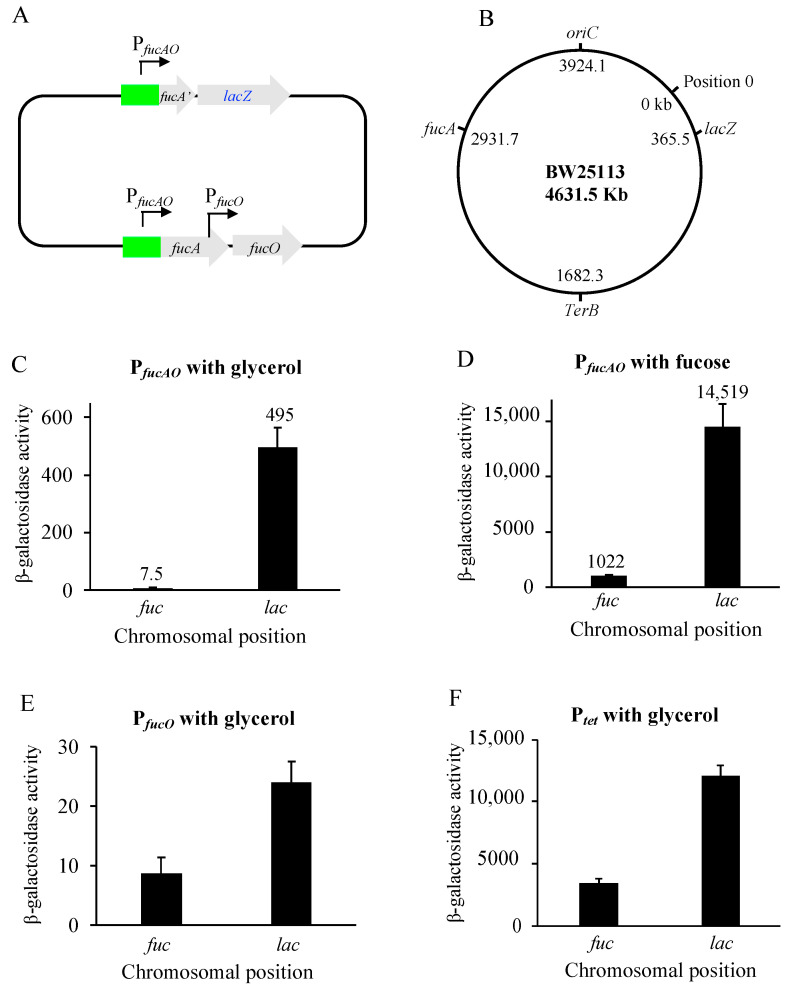
Chromosomal position effects on *fucAO* promoter activities. (**A**) Diagram showing P*_fucAO_* driving *lac*Z at the *lac* locus while the native *fucAO* operon is intact. *fucA*’ refers to the first 10 codons of *fucA*, followed by a stop codon. (**B**) Diagram showing the chromosomal positions for the *fucA* gene and the *lacZ* gene. The positions of these genes are given in kb with respect to Position 0. These two genes (*fucA* and *lacZ*) have similar distances from the *oriC* site. (**C**) The promoter activities of P*_fucAO_* at the *lac* locus and the *fuc* locus under noninducing conditions (*p* < 0.0001). (**D**) The promoter activities of P*_fucAO_* at the *lac* locus and the *fuc* locus under inducing conditions (*p* < 0.0001). (**E**) The promoter activities of P*_fucO_* at the *lac* locus and the *fuc* locus under noninducing conditions (*p* < 0.05). (**F**) The promoter activities of P*_tet_* at the *lac* locus and the *fuc* locus under noninducing conditions (*p* < 0.05).

**Figure 6 ijms-25-03946-f006:**
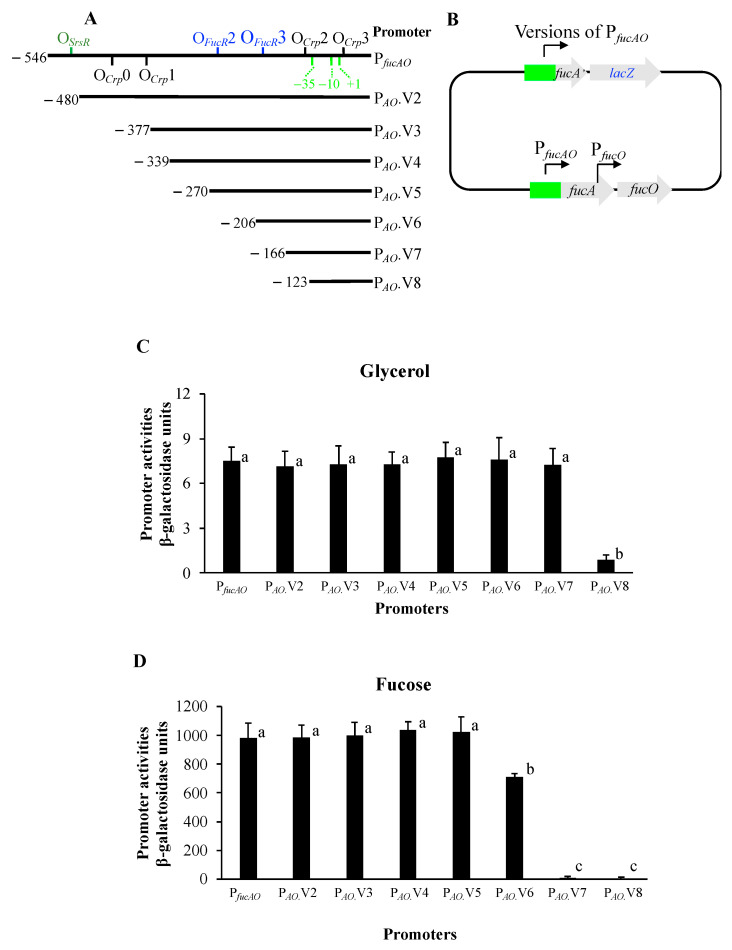
Determination of the minimal regulatory region needed for full *fucAO* promoter activity. (**A**) A schematic diagram showing various shorter regions used for promoter activity assays. The top line denotes the entire *fucPIK*/*fucAO* intergenic region. The “−1, −10 and −35” label in green refers to the newly identified P*_fucAO_*. Binding sites for Crp, FucR, and SrsR are labeled on the top line. The numbers on the left side of each line refer to the locations relative to the *fucA* start codon. (**B**) Diagram showing various versions of P*_fucAO_* driving *lac*Z expression at the *lac* locus while the native *fucAO* is intact. (**C**) The promoter activities of various versions of P*_fucAO_* under noninducing conditions (ANOVA *p* < 0.001). P*_AO_*.V8 exhibited significantly lower activity than all other promoters (all *p*-values < 0.05). The first seven promoters from the left have similar activities (all *p*-values > 0.05). (**D**) The promoter activities of various versions of P*_fucAO_* under inducing conditions (ANOVA, *p* < 0.0001). P*_AO_*.V7 and P*_AO_*.V8 had similar activities (*p* > 0.05). P*_AO_*.V6 had significantly higher activity than P*_AO_*.V7 (*p* < 0.01) and P*_AO_*.V8 (*p* < 0.01) but had significantly lower activity than the first six promoters from the left (all *p*-values < 0.05). The first five promoters on the left side have similar activities (all *p* > 0.05). For (**C**,**D**), the same letters denote no significant differences while different letters denote significant differences.

**Figure 7 ijms-25-03946-f007:**
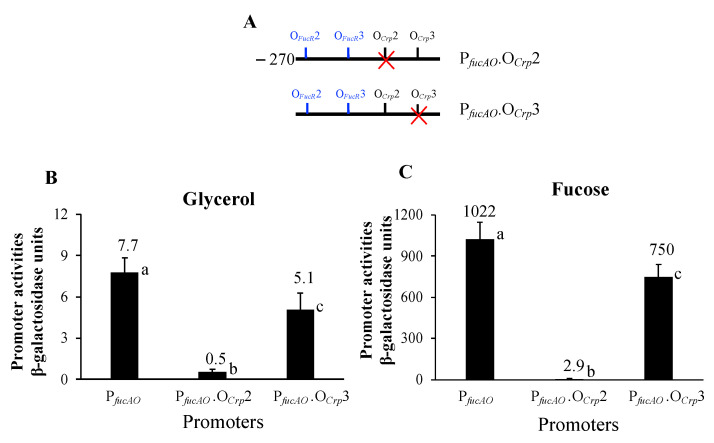
Effect of mutating Crp binding sites on P*_fucAO_* promoter activity. The minimal version P*_fucAO_*.V5 was used for these assays. The strains were cultured in M63 with either glycerol or fucose. (**A**) Diagram showing the mutations of the Crp binding sites. The binding site to be mutated is marked by a red cross. (**B**) The activities of P*_fucAO_* with altered O*_Crp_*2 or O*_Crp_*3 under noninducing conditions (ANOVA *p* < 0.0001). Different letters denote significant differences one another. P*_fucAO_*.O*_Crp_*2 had significantly lower activities than P*_fucAO_* (*p* < 0.0001) and P*_fucAO_*.O*_Crp_*3 (*p* < 0.001). P*_fucAO_*.O*_Crp_*3 had significantly lower activity than P*_fucAO_* (*p* < 0.01). (**C**) The activities of P*_fucAO_* with altered O*_Crp_*2 or O*_Crp_*3 under inducing conditions (ANOVA *p* < 0.0001). Different letters denote significant differences one another. P*_fucAO_*.O*_Crp_*2 had significantly lower activities than P*_fucAO_* (*p* < 0.0001) and P*_fucAO_*.O*_Crp_*3 (*p* < 0.01). P*_fucAO_*.O*_Crp_*3 had significantly lower activity than P*_fucAO_* (*p* < 0.05).

**Figure 8 ijms-25-03946-f008:**
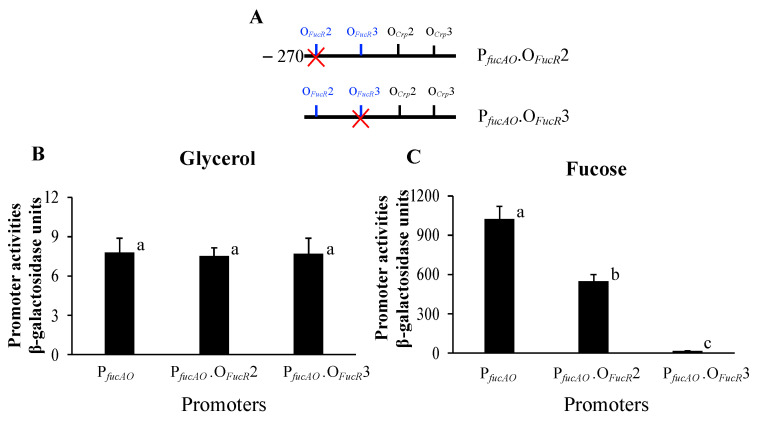
Effect of mutating FucR binding sites on P*_fucAO_* promoter activity. The minimal version P*_fucAO_*.V5 was used for these assays. The strains were cultured in M63 with either glycerol or fucose. (**A**) Diagram showing the mutations of FucR binding sites. The binding site to be mutated is marked by a red cross. (**B**) The activities of P*_fucAO_* with altered O*_FucR_*2 or O*_FucR_*3 under noninducing conditions (ANOVA *p* > 0.05). No significant activities (marked by the same letter a) were observed among these three promoters (all *p*-values > 0.05). (**C**) The activities of P*_fucAO_* with altered O*_FucR_*2 or O*_FucR_*3 under inducing conditions (ANOVA *p* < 0.001). Different letters denote significant differences one another. P*_fucAO_*.O*_FucR_*2 had significantly lower activity than P*_fucAO_* (*p* < 0.05) but had significantly higher activity than P*_fucAO_*.O*_FucR_*3 (*p* < 0.01), while P*_fucAO_*.O*_FucR_*3 had significantly lower activity than P*_fucAO_* (*p* < 0.001).

## Data Availability

The data presented in this study are available on request from the corresponding authors.

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
