# Peer review of "Comprehensive Characterization of fucAO Operon Activation in Escherichia coli"

_ijms, 2024, doi:10.3390/ijms25073946_

Round 1

Reviewer 1 Report

Comments and Suggestions for Authors

The manuscript describes a nice thorough classical study of the regulation of an E. coli metabolic operon. The main findings are the mapping of the transcript start site (TSS) for the primary operon promoter, demonstration that the operon is activated by cAMP-Crp and FucR protein, and the identification and verification of primary binding sites for these activators. The induction of the operon by FucR is unusually large and requires the presence of the primary Crp binding site. The most surprising finding is that the expression of the operon is strongly dependent on the location of the promoter, being more than 50 fold higher un-induced and 14 times higher fully induced when the promoter is relocated to the position of the lac operon (at the same distance from oriC on the opposite arm of the chromosome).

In addition to some minor points – see below – I only have two points of concern.

Firstly, I think the authors put too much emphasis on the presence of two binding sites for both Crp and FucR. The effect of mutating the secondary sites is very small (30 % and 50% respectively) compared to the massive effects of mutating the primary binding sites. The secondary Crp site is downstream of the TSS, and the effect seen could just as well be caused by changed mRNA structure influencing mRNA stability/translation efficiency. The authors have not shown that the effect is dependent on cAMP-CRP, they could easily have measured expression in glucose minimal medium in addition to the glycerol medium to substantiate that the reduction is cAMP dependent. If they do not confirm the cAMP dependence they should modify the text in results (lines 370-373) and especially in the discussion (lines 503-517 + 551).

Secondly I would like a more detailed discussion about the chromosomal location effect, including the strong probability that it is mediated by HNS – what is the AT % of the promoter region, are there H-NS “binding motifs”, a comparison with the bgl operon, the authors have studied “silencing” of the bgl operon by HNS and recently published papers concerning this. I am also missing a little discussion in relation to the old studies with activation of the fucAO operon by IS5 insertion (i.e. ref 17).

Minor points:

Figure 2 C: 2 columns are labelled DfucR, the third should be Dcrp.

Figure 2 E: I would like to have more digits shown on the Y-axis – to clearly show that it is a log10 scale.

Figure 3: the font is very small, making it hard to read.

Line 238: “feathers” should be “features”

Line 240: “to be the -10 element and -35 element respectively” change to “… -35 and the -10 elements” to fit with the order of the sequences given above- it is not all that like me knows these consensus sequences by heart

Line 248: “is derived for a” for should be “from”

Line 325: “FruR” should be FucR

Figure 6 A: the font for the O sites is very small and unclear due to the italics

Lines 397 and 406 : show the wt sequences together with the mutated (or remove the sequences and refer to 4.6 in M+M

Line 421 and 428: look at the language

Lines 456 – 461: confusing text rewrite, and this seems like some leftover from discussing why you doubted the promoter indicated in Ecocyc.

Line 570-71: lower and upper case is missing for chemical formulas and conc. of thiamine.

Line 645: “RT” is that room temperature?

Lines 697 and 707: you write that letters are in bold, but they are not

Line 728: “LacZ-Buffer” what is that ? is it the Z-buffer used in ref 60 the Miller 1972 book ?

Line 731: the ONPG solution – what concentration ?

Author Response

See attachment for our point-by-point responses to this review.

Reviewer 2 Report

Comments and Suggestions for Authors

This manuscript presents new information on the regulation of the fucose metabolism regulon. The promoter regions are experimentally analyzed and the role of different regulators is characterized. Even though the significance for a general audience may be somewhat limited, the work is scientifically sound and exhaustive. I only have some minor comments on the presentation of data:

- Figure 2. Panel B should indicate at what time/OD/growth phase was B-galactosidase activity measured. Panel C, there is a mistake in the X axis, the fucR mutant appears twice. Panel E, I recommend presenting the time in hours. Why were the other mutants not included here? It would be important to check their growth.

- Figure 4. What are the differences between panels A and B? Panel C is a bit confusing, since Crp sites are not presented until section 2.7.

- Along these lines, perhaps some rearrangement of the results could be considered. I would recommend moving section 2.5 to the last part of Results. The observation that the operon is in a low transcription region is interesting, but since it is not studied in detail, it seems less relevant for the rest of the data. That way, the characterization of the regulatory sites in the fucO promoter is presented in a logical sequence, and then the effect of the chromosomal location, as a further element to consider in the whole picture.

Author Response

(The authors gave the same response as above.)

Reviewer 3 Report

Comments and Suggestions for Authors

Manuscript: MDPI-IJMS-2894543

The authors studied comprehensive characterization of fucAO operon activation in Escherichia coli.

The authors presented fuc regulon and the L-1,2-propandiol pathway (Fig. 1), fucAO operon activities in wild type cell (Fig. 2), fucAO promoter activities (Fig. 3), determining the transcriptional TSS (Fig. 4), chromosomal position effects on fucAO promoter activities (Fig. 5), determination of the minimal regulatory region (Fig. 6), effect of mutating Crp binding sites (Fig. 7), effect of mutating FucR binding sites (Fig. 8). The authors concluded and reported that the data strongly indicate that FucR mainly functions to facilitate the binding of Crp to its upstream site, which in turn activates the fucAO promoter by efficiently recruiting RNA polymerase.

This is a well-designed and well-written manuscript. The authors have provided novel information in fucAO operon activation.

Author Response

Thank you for your positive review! We would not make any changes to our manuscript since no suggestions and comments are provided from this review.

Reviewer 4 Report

Comments and Suggestions for Authors

Authors present here in the manuscripte with title "Comprehensive Characterization of fucAO Operon Activation in Escherichia coli" a deep and fully documented study of the different promoters and their specific activation in the operon fucAO. The manuscript is well written and most of the assays are well connected; but along the text, there are references for future sections that make it confusing to read. However, some changes should be implemented to be accepted for publication, i.e., statistical analysis is missing as well as some experimental information. For that reason, my recommendation is a major revision.

Major comments

- Perform statistical analysis in: Figure 2B, C, D; Figure 3 D, F, G, I; Figure 4 A, B; Figure 5 C, D, E, F; Figure 6 C, D; Figure 7 B, C; Figure 8 B, C.

- Add the statistical analysis subsection in methods sections

- Line 153: do you mean bar? It is not needed to specify, since the graph is labeled.

- Lines: 198; 209, 211, 369: same comment than 153

- Line 219: remove see Discussion

- How many replicates are in each assay? The average of which "n" represents each bar? What do the error bars represent? Please specify in each footnote or in the methods sections.

Minor comments

- Use wild type, wildtype, or wild-type, not all forms

- Line 108: which strain?

- Line 130-131 seems not connected to the paragraph. Please rephrase.

- Line 171: (hypothetical promoter)

- Line 226: belongs to methods

- Chromatograms in Figure 4 are not needed and the quality of the image is not the best. Neither lines 230-231.

- Line 234:  σ70

- Line 238: features?

- Line 240: remove "See" 

- Line 436: remove the quotation mark from aerobically

- Line 491: remove see below

- Line 567: °C

- Line 697; 707: there is nothing bold

- Title in 716 (4.7.β-Galactosidase)

- Line 728: Do not start a sentence with a number (Two hundred)

- In general, remove all the "see Figure", "See in section X", ..., just referring to the figure/section is enough and more professional

Author Response

(The authors gave the same response as above.)

Round 2

Reviewer 4 Report

Comments and Suggestions for Authors

The authors have implemented most of my previous comments, and I consider this version more appropriate for publication, while some improvement is recommended.

I suggest clarifying the significance of the graphs since the chosen format with letters is not common and it is not explained in the footnotes, only the p-value threshold (what a, b, or c means?). Also, I'd recommend clarifying when statistics are run and significance is found, between which conditions. For example, in Fig 8C bars are labeled as a, b, and c; what a, b, and c mean? And when in the footnote it says "The activities of PfucAO with altered OFucR2 or OFucR3 under inducing conditions (p < 0.001)." b and c are same significance? Are they both compared with pfucAO, there is significance between R2 and R3? I found that confusing along the text.
